

# Large-scale water scarcity assessment under global changes: insights from a hydroeconomic framework

Noémie Neverre[1,2,3], Patrice Dumas[1,4], and Hypatia Nassopoulos[5]

[1]Centre International de Recherche sur l'Environnement et le Développement (CIRED), Nogent sur Marne, France
[2]Centre National de la Recherche Scientifique (CNRS), Paris, France
[3]Ecole Nationale des Ponts et Chaussées (ENPC), Champs sur Marne, France
[4]Centre de coopération International en Recherche Agronomique pour le Développement (CIRAD), Paris, France
[5]Ecole des Ingénieurs de la Ville de Paris (EIVP), Paris, France

*Correspondence to:* N. Neverre (neverre@centre-cired.fr)

**Abstract.**

Global changes are expected to exacerbate water scarcity issues in the Mediterranean region in the next decades. In this work, we investigate the impacts of reservoirs operation rules based on an economic criterion. We examine whether can they help reduce the costs of water scarcity, and whether they become more relevant under future climatic and socioeconomic conditions.

We develop an original hydroeconomic model able to compare water supply and demand on a large scale, while representing river basin heterogeneity.

On the supply side, we evaluate the impacts of climate change on water inflows to the reservoirs. On the demand side, we focus on the two main sectors of water use: irrigation and domestic sectors. Demands are projected in terms of both quantity and economic value. Coordinated operating rules of the reservoirs are set up, considering spatial and temporal trade-offs. The

10 objective is the maximisation of water benefits.

The methodology is applied to Algeria at the 2050 horizon. Our results show that the supply-demand imbalance and its costs will increase in most Algerian basins under future climatic and socioeconomic conditions. Our results suggest that the benefits of operating rules based on economic criteria are not unequivocally increased with global changes. In some basins the positive impact of economic prioritisation is higher in future conditions, but in other basins it is higher in historical conditions.

Given its generic nature and low data requirements, the developed framework could be implemented in other regions concerned with water scarcity, or extended to a global coverage.

*Keywords* Water scarcity costs, Large scale, Hydroeconomic modeling, Domestic water demand, Irrigation demand, Operating rules

## 1 Introduction

Climate change, demographic growth and economic development are expected to impact the water supply-demand balance and exacerbate water scarcity issues in the Mediterranean region in the next decades. In this context, there is a need for water



resources assessments to anticipate future water scarcity issues and their economic impacts. For water using sectors, water shortages mean unrealised benefits.

The Mediterranean region is well equipped with dams (Margat and Treyer, 2004). Man made reservoirs have an important impact on water fluxes (Biemans et al., 2011; Haddeland et al., 2014). They help regulate climatic variability in time and space, to distribute water when needed by demands. Moreover, when water is scarce reservoirs can help increase the water benefits, by allocating the available water to the most valuable uses. In hydroeconomic models, water allocation between competitive uses is based on the economic benefits they generate (Harou et al., 2009), and reservoirs can be managed with the objective of maximising the total economic benefits generated by water uses. Although economic rules are not often used in practice, water valuation could be used as a proxy for allocation policies, in the absence of precise information on the priorities set between the different demands in the different basins.

Taking into account the economic value of water enables not only to allocate water to the most valuable uses, but also to estimate the direct costs of water scarcity, in terms of unrealised economic benefits. To do so, a first step is to measure the benefits of water in its different uses, i.e. to determine the economic value of water (Young, 2005). Then assessing the missing quantity of water gives the corresponding unrealised economic benefits. Knowing water value in its different uses, it is possible to manage the available water so as to minimise the economic costs of scarcity.

The aim of this work is to examine if reservoir operation rules designed to maximise the economic benefits of the allocated water can help reduce the costs of future water scarcity under global changes in the Mediterranean.

Water resources assessments can be carried out at various spatial levels, from the catchment level to the global scale, with different levels of complexity for the water management infrastructure representation.

At the river basin scale, reservoirs can be represented as a network, with a nodal structure, and managed in a coordinated way. Some river basin level assessments cover extended geographic areas, as does the CALVIN model in California (Medellín-Azuara et al., 2008). But such models are developed for a specific basin or area, and use detailed data that are not generic. Some tools are developed to be flexible, and easily implemented to different basins, for instance the Water Evaluation And Planning system (WEAP) (Stockholm Environment Institute, 2011), but they also have high data requirements. In general, local scale approaches require a lot of data, which is not available at a large scale, and they are not applicable to regions where data is scarce.

In the global scale literature, the representation of reservoirs networks is challenging. Some studies do not represent reservoirs. The water resources assessment can be grid-based, without flow routing (Arnell, 2004). Some grid-based studies consider grid cell "storages", which are linked through flow routing (Oki et al., 2001). Other assessments take into account reservoirs, either without specific operation rules (Döll et al., 2003), or with reservoirs operating individually with generic rules (Hanasaki et al., 2008; Ward et al., 2010), or with optimised individual operation rules (Haddeland et al., 2007). In general, global scale studies lack to consider the nodal structure of reservoirs systems and the possibility of coordinated operation of reservoirs for a better supply-demand balance.

The economic dimension is taken into account at the river basin scale (Medellín-Azuara et al., 2008; Pulido-Velazquez et al., 2008). Many basin-scale hydroeconomic models were developed (Harou et al., 2009). However, in the large-scale literature



approaches are mostly quantitative (Alcamo et al., 2007; Hanasaki et al., 2013a, b; Schewe et al., 2014; Strzepek et al., 2013). Some large-scale studies incorporate an economic assessment, by considering the costs of water infrastructure adaptation to climate change (Hughes et al., 2010; Ward et al., 2010) necessary to meet projected demands. Some approaches introduce priorities between water uses, for instance Strzepek et al. (2013) use ranking rules (absolute priorities) between uses: domestic

and industrial uses are the first in priority, then come irrigation and livestock uses. But water valuation (Young, 2005) is absent from the large-scale literature.

In order to investigate water scarcity issues in the Mediterranean under global changes, and the benefits of reservoirs management rules based on economic criteria, we try to bridge the gap between existing large-scale and smaller-scale approaches. We develop an original generic hydroeconomic model able to compare future water supply and demand on a large scale, while

representing river basin heterogeneity. It takes into account man-made reservoirs and their coordinated operation, which relies on an evaluation of water economic benefits in the different water using sectors.

This paper first describes the modelling framework: it presents the methods for demand and supply projection, and for the reconstruction of the supply-demand network and the operation of the reservoirs. The framework is then applied to Algeria, at the 2050 horizon.

## 15  2  Projection of future inflows and demands

### 2.1  Runoff and flow accumulation

On the supply side, we evaluate the impacts of climate change on water availability, following the ODDYCCEIA[1] methodology (Portoghese et al., 2013). Water inflows to the reservoirs are computed at the monthly time step. Each reservoir's inflow corresponds to the summed runoff over the reservoir's upstream sub-basin, similarly to Islam et al. (2005). The sub-basin

flow-accumulation area of each reservoir is determined based on a Digital Elevation Model (HYDRO1k, 2009). Runoff data are taken from the grid cell outputs of the Centre National de Recherches Météorologiques (CNRM) climatic model (Dubois et al., 2012), which uses a stretched-grid global climate model zoomed on the Mediterranean coupled with a high resolution oceanic model of the Mediterranean, under the A1B IPCC-SRES emission scenario.

### 2.2  Projecting water demands and values

On the demand side, we focus on the two main sectors of water use: irrigation, which represents 65% of water uses in the Mediterranean basin and 58% in Algeria, and the domestic sector, which accounts for 13% of water uses in the Mediterranean basin and 27% in Algeria (Margat and Treyer, 2004). Irrigation and domestic demands are projected in terms of both quantity and economic value.

---

[1]ODDYCCEIA is the name of the framework. The abbreviation stands for Optimal Dam Dimensioning Yield and Climate Change Economic Impact Assessment. The acronym does not necessarily match the current use of the framework. It was coined for the Nassopoulos et al. (2012) paper, dedicated to cost benefit analysis and robust decision making for dam dimensioning adaptation under uncertain climate change. The ODDYCCEIA framework was also used to analyze imbalances between water supply and irrigation demand in the Mediterranean basin under climate change (Portoghese et al., 2013).





### 2.2.1 Irrigation sector

Irrigation water demand is projected under climate change (A1B scenario), for twelve different types of crops, at the 0.5°
spatial resolution.

Globally available data on irrigated areas and crops are combined in order to determine irrigated crops localisation. Then,
crops irrigation requirements are computed as the difference between potential crop evapotranspiration (ETc) and usable pre-
cipitation, for the different stages of the growing season, following the Allen method (Allen et al., 1998), as in the ODDYC-
CEIA framework (Portoghese et al., 2013).

The economic value of irrigation water is calculated using a yield comparison approach between rainfed and irrigated
crops: additional profits made possible by irrigation are compared to its additional costs, and the value of water consists of
the additional value added associated with the use of water. In order to estimate irrigated and rainfed yields under future
hydro-climatic conditions, we model yield as a simple function of usable precipitation and ETP.

Further details on the methodology are available in Appendix A.

### 2.2.2 Domestic sector

We project the combined effects of demographic growth, economic development and water cost evolution on future domestic
demands. Our methodology is to build three-part inverse demand functions, at the country scale (Neverre and Dumas, 2015).
The economic value of domestic water is defined as the economic surplus (i.e. difference between the marginal willingness to
pay for water and the cost of water along the demand curve). Further details on the methodology are available in Appendix B.

Projected demands are then spatially distributed. Current urban areas localisation and population distribution are taken from
the GRUMP database (Center for International Earth Science Information Network (CIESIN) et al., 2004). We assume that
the location of future urban areas remains the same as present, and that future population growth is homogeneously distributed
among existing locations (the population ratio of each city over the total population remains unchanged).

## 3 Reconstruction of the network

Information on the physical links between reservoirs and demands are not available at large scale, the network has to be
reconstructed.

### 3.1 Demand-reservoir association

Reservoirs are located using Aquastat (AQUASTAT Program, 2007), and the reservoirs network is reconstructed by defining
upstream-downstream links.

A generic methodology was developed for the determination of demand-reservoir links (Portoghese et al., 2013). Links are
reconstructed based on a topological cost constraint, with a penalisation of the distance covered and uphill moves along the
potential supply-demand paths. The paths start from the stream that flows down from the reservoir, not from the reservoir itself.





For a reservoir $r$ and a demand $i$, the cost of the $r, i$ link is:

$$cost_{r,i} = \min_{path} \left( d_{r,i}^{path} + 10^4 \cdot H_{r,i}^{path} \right) \tag{1}$$

with $d_{r,i}^{path}$ the distance covered along a supply-demand path and $H_{r,i}^{path}$ the altitude differential of uphill movements along the path. A penalty coefficient of $10^4$ is attributed to uphill movements, implying that going up one meter is $10^4$ times more costly than covering one meter horizontally. The path corresponding to the minimum cost is the supply-demand path for this reservoir-demand couple $r, i$. Each demand is associated to only one reservoir.

### 3.2 Order of the demands on a stream

Water demands are not entirely consumptive (Table 1), they generate return flows that may help satisfy downstream demands. In order to take into account return flows, it is necessary to know the order of the demands water intakes on the stream.

First, we determine the point of intersection between the final supply-demand path and the stream. This is the potential location of the demand's inlet.

Demands located close to the stream are likely to have their own water intake. However, demands located far from the stream are likely to share common supply infrastructure (pipes, channels, aqueducts, etc.), and share a common water intake on the stream. Therefore, in a second step, we group potential inlets based on the average topological cost of the supply-demand paths for the considered stream. We assume that if the conveyance of water does not necessitate uphill moves of more than 10 meters, then water intakes can be numerous, and we do not regroup them. They can be located as close as 1 km from each other, which is the resolution of the Digital Elevation Model. If the supply-demand path requires uphill moves higher than 160 meters (or covers distances longer than 1600 km), water intakes are grouped so that they cannot be located closer than 21 km from each other. In between, the spacing between water intakes increases proportionally to the topological cost, by steps of 5 km.

### 4 Operation of the reservoirs

Once the demand-reservoir network is reconstructed, the next step is to compute coordinated operating rules for the reservoirs in each river basin, taking into account available inflows and potential demands.

Multiple reservoirs systems operation has been extensively studied. Simulation models, which require the prior specification of operating rules (Oliveira and Loucks, 1997), can be used. Defining effective predefined operating rules is a challenge for complex multi-reservoirs systems, and at large scale it is not possible to use operator defined rules. A wide range of optimisation techniques exist (Labadie, 2004); optimisation can be used to help define the parameters of operating rules, and combined simulation-optimisation approaches were developed (Rani and Moreira, 2009).

Our approach is to build something not too complex, which is not too data-intensive nor computationally-intensive. We use a parameterisation-simulation-optimisation (PSO) approach (Nalbantis and Koutsoyiannis, 1997; Koutsoyiannis and Economou, 2003), generalised to more complex reservoirs systems, and with prudential rules.



## 4.1 Objective function

Operating rules are based on the maximisation of water benefits, over time and space. The objective function is $Max(B_{tot})$, where:

$$B_{tot} = \sum_{t=1}^{T} \sum_{r=1}^{R} \sum_{n=1}^{N_r} D_{n,t} \cdot v_{n,t} \tag{2}$$

$T$ is the number of time periods, $R$ is the number of reservoirs in the network (i.e. also the number of streams), and $N_r$ the number of demands on the stream just downstream of reservoir $r$. $D$ is the satisfied demand, and $v_{n,t}$ is the value of water for demand $n$ on month $t$ (per unit of water).

This objective affects the water management rules on two levels: $i$) the allocation of water between demands on one stream, $ii$) the coordinated management of the whole reservoirs' system.

## 4.2 Water allocation between demands on one stream

In order to reduce computation time, we group demands based on their valorisation of water. First, given total yearly demand on the stream, we define a number of value classes. The larger the demand, the more the classes: for each additional 10 million m$^3$ (i.e. roughly the annual water use of a city of 100 thousand inhabitants), we consider one more class. We then determine the value bounds of the classes so that the total cumulated water benefits in each class is identical. Finally, for each inlet, we group demands pertaining to the same value class, and compute the monthly total demand and average value of this aggregated demand. This is done separately for irrigation and domestic demands. Water is then allocated considering these aggregated demands.

For each month, given a release $I_0$ for the stream (released from the sub-system of upstream reservoirs), we determine the satisfied demands $D_n$ ($n \in (1,...,N)$) under the objective:

$$Max \sum_{n=1}^{N} D_n v_n \tag{3}$$

Subject to the following continuity constraints, where $l$ indexes inlets, and inlet $l+1$ is downstream from inlet $l$:

$$I_l = I_{l-1} - \sum_{n_l=1}^{N_l} \gamma_{n_l} D_{n_l} \; ; \quad \sum_{n_l=1}^{N_l} D_{n_l} \leq \min\left(\sum_{n \; in \; l} M_{n_l}, I_{n-1}\right) \tag{4}$$

$N_l$ is the number of demands located on inlet $l$. $M_{n_l}$ is the potential demand, $\gamma_{n_l}$ the consumptive ratio, $v_{n_l}$ the water value, and $D_{n_l}$ the satisfied demand. $I_0$ is the inflow entering the stream, $I_l$ is the inflow downstream inlet $l$.

We suppose that return flows from a demand located on inlet $l$ are available for downstream demands at inlet $l+1$, without considering any decrease in quality.

The water allocation method within a stream gives priority to the high value uses and little consumptive uses.





### 4.3 Building coordinated operating rules for the reservoirs

For each stream of the network, at each time step, the operating rules of the reservoirs answer two questions: $i$) how much water to release from the total storage of upstream reservoirs for the demands of this stream, $ii$) how to distribute this release between the different reservoirs of the upstream sub-system. In order to manage water at best, these operation decisions should

be coordinated, for all reservoirs of the network.

#### 4.3.1 A parameterisation-simulation-optimisation approach

We use a PSO approach to set up operating rules, with two parameters for each node ($\alpha$ and $\beta$) to parameterise the choice between upstream branches (Nalbantis and Koutsoyiannis, 1997) and a prudential rule, as described below. Parameters are optimised using a genetic algorithm (Hınçal et al., 2011). This heuristic programming method may help avoid getting stuck in

local optimums (Labadie, 2004).

#### 4.3.2 Determining how much water to release for a given stream

We want to give priority to the satisfaction of demands with high valorisations of water and low consumptive rates. Thus, it can be preferable not to satisfy entirely a demand, if it enables the satisfaction of demands of a higher value, which can be located on another stream downstream, or occurring at a later time-period.

To take into account these spatial and temporal trade-offs, we introduce prudential parameters. A hedging rule (Draper and Lund, 2004) is used, to determine how much water to release for the demands of a stream, and how much water to retain for potential higher value uses. To avoid increasing computation time, and avoid overfitting, we use a one-point hedging rule (Draper and Lund, 2004), as illustrated in Figure 1.

First, we determine what minimum release would be necessary to satisfy all demands of the stream. This is the target release

$T$. In a second step, we determine actual release, based on the hedging parameter for the stream ($\alpha$). Under a standard operating policy (SOP), the reservoir would release water depending on the quantity available and the target release $T$: if the available water quantity is lower than $T$, all available water is released; if more than $T$ is available, the quantity $T$ is released; if there is more water available than $T$ plus what can be stored, the excess water is spilled and release consists of $T$ plus the spill (Figure 1). Under the hedging rule, when the available water quantity is lower than the trigger volume $V_{lim}$, there is some rationing:

release is lower than under the SOP, following the slope $\alpha$ (with $\alpha \leq 1$).

#### 4.3.3 Distributing release between reservoirs of the upstream storage sub-system

For reservoirs in series, a rule proposed by Lund and Guzman (1999) is used. When satisfying a demand, water is first extracted from the most downstream reservoir, then progressively from the upstream reservoirs. The objective is to leave the water as upstream as possible, where it will be available for a wider geographical area.

For reservoirs in parallel, we have to decide from which branch to withdraw the water allocated to the downstream demand. The parametric rule of Nalbantis and Koutsoyiannis (1997) is used. A $\beta_r$ parameter is defined for each reservoir $r$ upstream





of an node. With $N$ upstream reservoirs: $\sum_{r=1}^{N} \beta_r = 1$. The $\beta$ parameters determine the distribution of the empty storage space amongst the reservoirs in parallel, as a function of their respective storage capacities and common downstream demand (Portoghese et al., 2013). Details on the methodology are available in Appendix C.

### 4.3.4 Tree traversal

The coordination of operations throughout the demand-reservoir network is implemented using tree traversal, instead of a system of constraints. Two traversals interlock: one progresses downstream, to compute the water release for the demands of each stream; the other one progresses upstream, to distribute this release between reservoirs of the upstream storage sub-system (Appendix D).

## 5 Application to Algeria

As an illustration, the methodology is applied to Algeria. Future demand and supply are projected at the 2050 horizon. Year 2000 is the year of reference for the historical period. The model is run for fifty climatic years, centered around the year of reference: 1975-2025 for the historical period, and 2025-2075 for the future period.

### 5.1 Scenarios

#### 5.1.1 Demand projection

We project domestic demand under the Medium variant population scenario of UNO (UN, 2009), and SSP2 (Rozenberg et al., 2014) GDP evolution scenario. Domestic water cost is assumed to converge towards the cost of water in France, which is used as a proxy for the cost of water in a mature domestic water distribution and sewerage service. Cost-recovery ratio is assumed to converge towards one, following GDP per capita evolution (Neverre and Dumas, 2015).

Irrigation demands and values are projected under the assumption that crops growing periods are of fixed length, and with 20 no evolution of crops prices in the future.

The demand projection methods (Section 2.2) estimate on-site demands. To get corresponding withdrawals, we have to take into account distribution losses. Efficiency ratios are based on Margat and Treyer (2004). For Algeria, both irrigation and domestic sectors have a demand to withdrawal ratio of 50%. We assume that these ratios remain unchanged in the future.

#### 5.1.2 Supply

When runoff exceeds the reservoir's storage capacity, the excess water is spilled. Runoff and demands are computed at a monthly time step. The spill can be handled in two different ways. On the one hand, we can compute the spill at the beginning of each month, before computing demand satisfaction. In this case, spilled water that is not collected by downstream reservoirs is lost, it does not help satisfy demands. This is a pessimistic scenario, compatible with heavy concentrated rain. On the other hand, the spill can be computed while allocating water to the demands. In this way, all runoff participates in the satisfaction of





demands. This is an optimistic scenario, compatible with well distributed precipitations during the month. Both scenarios are compared, they are noted "spB" when spill is computed before demand satisfaction, and "spA" when spill is computed along with demand satisfaction.

We also compare the results of two operating rules strategies. Under the option "V⁺H⁺", demands are prioritised: the value

of water in its different uses is taken into account (V⁺) and one-point hedging is implemented (H⁺). Under the option "V⁻H⁻", the value of water is not taken into account (V⁻), and no hedging is implemented (H⁻).

## 5.2 Results

Results of satisfied demands and satisfied economic benefits are displayed in Table 2, under past and future conditions, under the V⁺H⁺ operating rules option. Results are presented for each river basin (Figure 7). Knowing that we model potential

demands, and that not all areas equipped for irrigation are actually irrigated (Benmouffok, 2004), we expect that the reservoirs may not be able to satisfy the whole demand.

The modelled historical rates of demand satisfaction seem pertinent for river basin 1186, which corresponds to the Chelif basin and the largest reservoirs system in Algeria, with 18 nodes (Figure 7). Results also seem appropriate for river basins 20, 35, 1191 and 1192, which correspond to smaller coastal basins. Modelled past demand satisfaction rates may be too high for

river basins 13 and 28 (small basins with only one reservoir).

For other river basins (1170, 1171, 1178, 1181, 33, 1190, 9 and 1189), our results display very low demand satisfaction rates in historical conditions. One reason may be that water supply sources other than those accounted for in our framework are used: groundwater (basins 1170, 1171 and 9), desalination (1178), small dams (9). Reservoirs can be erroneously associated to irrigation perimeters using other water supply sources (basins 33, 1981, 1189 and 1190). These issues are detailed in Appendix

F.

In most basins, the spill computation option has no impact (Table 2). In others, the "optimistic" spill option (SpA) increases demand satisfaction rate: +4.6-27% in past conditions for 1178, 1186, 1190 and 1192, and +2.2-5.5% in the endorheic basins 1170 and 1171.

Under future climatic and socioeconomic conditions, few basins (35 and 1171) experience an improvement in the demand-

25 supply balance. Their increase in satisfaction rates ranges from +3.5-12.6 percent points in demand satisfaction, and +0.2-5.6 percent points in terms of economic benefits satisfaction (Table 3). Most basins undergo a decrease in satisfaction rates under future conditions: up to -41.6 percent points in demand satisfaction, and -34.4 percent points in economic benefits satisfaction. Basin 28 is particularly affected, with a 91 percent points decrease in demand satisfaction.

Table 4 illustrates the impacts of demand prioritisation. In terms of water quantities, demand satisfaction can be higher

with prioritisation than without it (e.g. basin 1191), probably because water is allocated to less consumptive uses that generate more return flows. Demand satisfaction can also be lower with prioritisation (e.g. basin 1178 in past conditions), because when there is no hedging more water can be allocated to demands. The impact of demand prioritisation on economic value satisfaction rate is always positive: up to +27.1% in past conditions, and up to +22.5% in future conditions. For some basin (9, 33, 1170, 1189, 1190, 1191 and 1192), the positive impact of prioritisation is more pronounced under past conditions than





under future conditions. For other basins on the contrary (13, 20 and 28), the positive impact of prioritisation increases in the future, when these basins experience more pressure on the resource (Table 2). For the remaining basins, past and future benefits of prioritisation are comparable.

# 6    Discussion and conclusion

The developed methodology models domestic and irrigation water demands, as a function of socioeconomic and climatic conditions. It reconstructs water infrastructure networks and compares potential withdrawals to available water supplies, with a multiple-basin coverage. Simple parametric operating rules are implemented to manage the reservoirs in a coordinated way, for a better valorisation of water.

Overall, our results show that the supply-demand imbalance will increase in most Algerian basins in the future, under the
simulated socioeconomic and climate scenario.

Under future conditions, in some basins demand satisfaction (in terms of value) can be increased by up to 22.5% when using economic criteria to determine reservoirs operation rules. It suggests that global changes might be an incentive to use valuation in operating rules in these basins. In other basins, the benefits of reservoirs management based on economic criteria are less pronounced. In this case, trade-offs could arise between implementing economic based operation policies or not. Implementing
economic based priorities between uses may be complicated, costly, and involves acceptability issues. These difficulties should be compared to the expected gains in water benefits. The comparison of the benefits of operating rules based on economic criteria under historical and future conditions suggest that these benefits are not unequivocally increased with global changes.

Our approach combines a large-scale coverage with a representation of heterogeneities at the river basin level (climate, water infrastructure, human activities, etc.). This double focus is useful to assess operating rules or policies effects on a multi-basin
scale, as in the present paper. It would also be useful to investigate issues that occur on a large scale, for instance virtual water trade through markets of goods requiring water for their production. Being able to represent contrasted situations between basins, some suffering from water scarcity more than others, makes it possible to consider the water scarcity issue from a broader perspective than the usual water basin management level and consider possible interactions between basins. Modelling the economic benefits associated with water use, and the economic constraints associated with water shortage, is particularly
important to address such inter-basins issues, to understand how interactions may be fostered.

The generic nature of the framework, necessary to maintain this double scale focus, has its limits. In particular, there can be non-negligible errors in reservoirs-demands network reconstruction when using only globally available data. Some degree of validation seems to be needed for a closer look at basin-scale results. The framework is not designed to provide a detailed representation of catchments for operational purposes, but rather to represent localised impacts of global changes, with an
extended geographic coverage.

Validating the representation of reservoirs' operation policy would require data on naturalised and non-naturalised flows. Some data may be available for the Mediterranean (Ludwig et al., 2009), but their coverage, both in terms of years and locations, is far from complete. Some information on which demands suffer restrictions would also be needed, but to our





knowledge such data is not available. Besides, the purpose of the framework is not necessarily to reproduce observations. The operation rules built should perform better than uncontrolled flows, even if they do not match observations well. Since using explicitly economic value for water management is rare in practice, observations could also enable to evaluate if using water value as an allocation criteria is better than not to reproduce existing practices.

Other sectors of water use could be taken into account in the framework, such as electricity production (cooling, hydropower) or environmental flows. The framework's large scale would be particularly pertinent to consider the electricity sector, since electricity markets are of a large scale. The relevance of using water for producing electricity and the relative benefits of different production technologies could be investigated, depending on the price of electricity, as well as the plausibility of future energy mixes with regard to water availability.

It would also be relevant to incorporate groundwater into the framework. The application to Algeria highlighted that some areas rely on groundwater as a complementary or major water supply source. Groundwater management is often decentralised. An economic approach comparing water pumping costs to the economic benefits of the water demands could be used; the type of prioritisation and prudential rules we developed for surface water could also be generalised to aquifers.

    The developed framework is a first attempt at bridging the gap between global-scale and local-scale modelling approaches:
it offers the possibility of taking into account coordinated operation of reservoirs and economic valuation at large scale. Given its generic nature and low data requirements, it could be implemented in other regions concerned with water scarcity and its costs, or extended to a global coverage.

*Author contributions.* P.D. and N.N. designed the research and developed the methodology. All coauthors participated in developing the model code. H.N. developed the methodology for irrigation demand computation, and performed the validation of reservoirs-demands net-
work reconstruction. N.N. produced the results and wrote the manuscript.

*Acknowledgements.* The authors would like to thank Ankur Shah Delight for his input, and the Direction Générale de l'Armement for its financial support through a PhD grant.

## Appendix A:  Irrigation water demand projection and valuation

### Irrigation demand projection

Irrigation demand projection follows the methodology of ODDYCCEIA model (Portoghese et al., 2013), which analysed imbalances between water supply and irrigation demand in the Mediterranean basin under climate change.

    Historical irrigated areas are determined from globally available data on irrigated areas and crops (Siebert et al., 2005), and crop mixes in the different irrigation perimeters are taken from Agro-MAPS database (FAO, 2005). Future crops surfaces and types are assumed to be the same as in historical conditions (year 2000): we do not model changes in crop types distribution nor in areas equipped for irrigation.

Irrigation requirements are defined as the deficit between the potential crop evapotranspiration and the effective precipitation. Effective precipitation is computed following Döll et al. (2003). Crop evapotranspiration is computed for the different stages of the growing season us-





ing Allen et al. (1998) method. AQUASTAT Program (2007) is used for crop calendars, and growth phases are assumed to remain of the same duration in the future. The reference evapotranspiration ($ET_0$) is computed following the Hargreaves method. Then crop evapotranspiration ($ETc$) is determined using crop coefficients ($Kc$) values from Allen et al. (1998): $ETc = ET_0 \cdot Kc$.

Future irrigation needs are affected by climate change. Climatic data are taken from the CNRM climatic model (Dubois et al., 2012) outputs, using the A1B IPCC-SRES emission scenario.

Sixteen types of crops are differentiated in the ODDYCCEIA methodology: cotton, fodder, fruits, maize, oil-seed, oil-tree, potatoes, pulses, rice, rubber, sorghum, sugar beet, sugarcane, tobacco, vegetables and wheat.

Since livestock water use is much smaller than irrigation water use (Alcamo et al., 2007; Hanasaki et al., 2013a), in the present paper we consider only irrigation water needs.

## Irrigation water value

We estimate irrigation water value based on a "yield comparison approach" (Turner, 2004), a simple approach derived from the residual method (Young, 2005), in which respective costs and benefits of rainfed and irrigated production are compared. For a given crop in a given location, the additional profits made possible by irrigation are compared to its additional costs, and the value of water consists in the additional net benefits associated with the use of water.

We compute the value of water for each ODDYCCEIA crop type, in each irrigation perimeter location (i.e. at the $0.5°$ per $0.5°$ grid cell scale), as follows:

$$V = (B_{ir} - B_{rf})/W$$

Where $V$ is the volumetric value of irrigation water (in US\$/m$^3$). $B_{ir}$ is the net benefit obtained by the irrigated production of a given crop in a given location (in US\$/ha), and $B_{rf}$ is the net benefit that would be obtained if this crop was rainfed. $W$ is the quantity of water used for irrigating the crop in this irrigation perimeter (in m$^3$/ha). $V$ can be negative if the additional profits generated by irrigation do not offset its additional costs. In this case, rainfed production is preferable to irrigated production.

In order to be able to determine irrigated and rainfed yields ($Y_{ir}$ and $Y_{rf}$) in future hydro-climatic conditions, we model yield as a simple function of usable water and ETc, as described in Figure 2.

The simple piecewise linear yield function is calibrated for each crop type by means of these two points of reference: the couples (yield of reference, usable water to ETc ratio of reference), for rainfed and irrigated crops. Hence, for each crop type in each irrigation perimeter location, we have to determine $i$) historical usable water-to-ETc ratios, and $ii$) historical rainfed and irrigated yields.

For rainfed crops, historical precipitation-to-ETc ratios are computed at the $0.5°$ per $0.5°$ spatial resolution, based on average precipitation and ETc outputs of the CNRM climatic model (Dubois et al., 2012) calculated over fifty past climatic years. For irrigated crops, by construction, usable water-to-ETc ratio is always equal to one.

For available crop types, historical yields of reference are based on localised potential irrigated and rainfed yields from LPJmL (Bondeau et al., 2007). For other crops, they are based on data from FAOSTAT (FAOSTAT, 2013) and simple assumptions on yield ratios: we use data on the country scale average rainfed yield for each crop, and then assume that in each grid cell the average rainfed yield to potential yield ratio is equal to the precipitation to ETc ratio.

To take into account future yield increases associated with an increased use of other inputs, we add a yield change multiplier. Its value is taken from Alexandratos and Bruinsma (2012) data, at the 2050 horizon.





As with the residual method, the methodology results in the estimation of average values. For each crop in each location we can obtain a past value of irrigation water based on historical yields and climatic conditions, and a future value of irrigation water based on future climatic conditions and yield change multipliers, computed over fifty climatic years.

## Appendix B:  Domestic water demand projection and valuation

Given the low data availability when working at a large scale, we developed a framework to build generic demand functions (Neverre and Dumas, 2015).

Our approach is to build simple three-part inverse demand functions (Figure 3): the first part consists of basic water requirements for consumption and hygiene, which are very highly valued (e.g. drinking water); then the second part consists of additional hygiene and less essential uses, which are less valued than those of the first part (e.g. regular laundry); finally the third part consists of further indoor uses and
outdoor uses which are the least valued (e.g. lawn watering).

Then, we allow for the structure of our demand function to evolve over time, in order to take into account the effect of economic development on demand (Figure 3). As GDP per capita increases, households get more water using appliances (e.g. washing machines) and demand more water, until they eventually reach equipment saturation and water use stabilises even if income continues to grow. This economic development effect was modelled in the WaterGAP model (Alcamo et al., 2003). We use a similar methodology, but incorporate it into our
economic demand function framework. The size of the blocks of the demand function are scaled by economic development. The width of the second and third blocks grow with the level of GDP per capita, as illustrated in Figure 3. We assume that basic needs are not affected by this equipment effect, and the width of the first block is assumed fixed and based on literature (Howard and Bartram, 2003; Gleick, 1996).

Willingness to pay is estimated for some demand points of reference, based on econometric studies (Nauges and Thomas, 2000; Schleich and Hillenbrand, 2009; Frondel and Messner, 2008) or price data (domestic water prices, bottled water prices), then interpolated linearly to
form the slopes of the blocks.

Demand can be projected for a given year $t$ following these steps: first, determine $Q_{int}$ and $Q_{tot}$ depending on the level of GDP per capita on year $t$. This enables to build the economic demand function for that year $t$. Second, project price for year $t$, and determine the level of demand for that level of price based on the economic demand function (Figure 4). Then the value of water can be computed: it consists of the economic surplus, i.e. the difference between willingness to pay and the actual cost of water, (Figure 4).
The domestic demand functions we model account for both household uses, and commercial and collective uses.

## Appendix C:  Distributing release between reservoirs in parallel

Here we consider a system of reservoirs formed by $nl$ upstream reservoirs $l$ in parallel and a direct downstream reservoir $dp$. First, the reservoir $dp$ will cover, partly or totally, the downstream release. The residual release $\Delta R^0_{dp,t}$ will be covered by upstream reservoirs using the parametric rule (Nalbantis and Koutsoyiannis, 1997). The upstream system total final water volume is:

$$V^{syst}_{dp,t} = \sum_{l=1}^{nl} Vcur^0_{l,t} - \Delta R^0_{dp,t}$$





The parametric rule is used to determine from which upstream reservoir water is extracted. The target volume $S_{l,t}^{targ}$ of each upstream reservoir is:

$$S_{l,t}^{targ} = K_l + \beta_l \times \left( V_{dp,t}^{syst} - \sum_{l=1}^{nl} K_l \right) \qquad \text{(C1)}$$

These target volumes may not be consistent with constraints on volumes, i.e. volumes may be negative or exceed storage capacity. Hence,
their corrected target volumes $S_{l,t}^{targ,corr}$ are:

$$S_{l,t}^{targ,corr} = \begin{cases} K_l, & \text{if } S_{l,t}^{targ} > K_l, \\ 0, & \text{if } S_{l,t}^{targ} < 0, \\ S_{l,t}^{targ}, & \text{otherwise.} \end{cases} \qquad \text{(C2)}$$

Once these inconsistencies are corrected, the new target volumes may not add up to the system's volume $V_{dp,t}^{syst}$. Hence for the second change, a correction coefficient is defined (Nalbantis and Koutsoyiannis, 1997). The correction factor $\phi_{dp,t}$ is computed as follows:

$$\phi_{dp,t} = \frac{(V_{dp,t}^{syst} - \sum_1^{nl} S_{l,t}^{targ,corr}) \times K_l}{\sum_1^{nl} S_{l,t}^{targ,corr} \times (K_l - S_{l,t}^{targ,corr})} \qquad \text{(C3)}$$

Based on the $\phi_{dp,t}$ correction factor, new target volumes $S_l^{targ,new}$ are computed as:

$$S_{l,t}^{targ,new} = S_{l,t}^{targ,corr} \times \left\{ 1 + \left[ \phi_{dp,t} \times \left( 1 - \left( \frac{S_{l,t}^{targ,corr}}{K_l} \right) \right) \right] \right\} \qquad \text{(C4)}$$

These two corrections have to be done repeatedly until the previous constraints are fulfilled.

$S_{l,t}^{targ,new}$ can be greater than the current volume $Vcur_{l,t}^0$. For the set $R^g$ of upstream reservoirs for which this condition holds ($R^g = \{r/S_{r,t}^{targ,new} > Vcur_{r,t}^0\}$), the final target volumes is set equal to their current volumes (Nassopoulos, 2012). The residual release is cov-
ered by the set $R^l$ of remaining upstream reservoirs which have target volumes lower than their current volumes ($R^l = \{r/S_{r,t}^{targ,new} < Vcur_{r,t}^0\}$). Each one of these reservoirs contribution is set to be proportional to its current volume, and its final target volume will be $\forall l \in R^l$:

$$S_{l,t}^{targ,fin} = \frac{Vcur_{l,t}^0}{\sum_{r \in R^l} Vcur_{r,t}^0} \times \left( V_{dp,t}^{syst} - \sum_{r \in R^g} Vcur_{r,t}^0 \right) \qquad \text{(C5)}$$

## Appendix D: Tree traversals

The coordination of operations throughout the demand-reservoir network is implemented using tree traversal, instead of a system of constraints (Portoghese et al., 2013). Two traversals interlock: one progresses downstream, the other one upstream.

For the traversal of the whole network, we start from the most upriver parts and progress downstream (Figure 5). This downstream traversal determines how much water is released for the demands of each stream, based on the hedging rule. Once a stream is processed, we move downstream, and the reservoirs of the upstream system already processed are aggregated into one reservoir. Going further down, the hedging





rules are used for the satisfaction of the demands and the processed streams are aggregated, in the same way. This recursion continues until reaching the network root.

At the same time, the network is traversed upwards. Each time a demand has to be covered by an upstream reservoirs aggregate, we must determine which reservoir of each upstream aggregate will contribute to the release. For this purpose, the rules to distribute release between

reservoirs in series or in parallel are used, while progressing upwards (Figure 6). This recursion continues until reaching leaf reservoirs.

## Appendix E:  Location of Algerian reservoirs systems

Cf. map in Figure 7.

## Appendix F:  Validation of reservoirs-demands links reconstruction in Algeria

For some river basins (1170, 1171, 1178, 1181, 33, 1190, 9 and 1189), our results display very low demand satisfaction rates in historical

conditions (Table 2).

River basins 1170 and 1171 are endorheic basins located in arid areas (Figure 7). It is not surprising that the modelled supply in the reservoirs system does not meet the modelled demand. Supply sources other than dams are used in these areas, notably the Saharan aquifer.

Basin 1178 is associated to the city of Oran, the second largest city in Algeria. However, Oran suffered from water deficits and had to resort to desalination as an alternative water supply source. This can explain the low demand satisfaction rate modelled for this reservoirs

system.

Errors in reservoirs-demands network reconstruction can explain the demand-supply imbalance in the remaining problematic reservoirs systems. In basin 1181, the reservoirs are associated to demands from large irrigation perimeters in Morocco. These associations are probably erroneous, and the total demand associated to the reservoirs system is too large. In basin 33, the reservoir is associated to irrigation demands from the Mila province. However, the reservoir purpose is domestic supply, and Mila province irrigation perimeters are not supplied by

reservoirs (Messahel et al., 2005). Basin 1190 is also associated to irrigation perimeters, although it is reported to supply water only to cities (Ministère des Ressources en Eau Algérien, 2007). Basins 9 and 1189 are located close to each other. There are multiple problems with the reservoirs-demands network reconstruction in this area. The two reservoirs of 1189 are incorrectly located in the database we used (AQUASTAT Program, 2007), which can alter inflow computation and supply-demand paths computation. Basin 9 is associated to the city of Alger, the largest Algerian city, whereas Alger should be associated to closer and smaller dams, that are missing in the dams database

used for the modelling (AQUASTAT Program, 2007). There is also one reservoir missing in the Aquastat database, the Boukerdane reservoir, which should supply water to the irrigation perimeters of Sahel Algérois Ouest. The model associates these irrigation perimeters to the other reservoirs, and their associated demand is therefore higher than it should be. Reservoirs of basin 9 are also erroneously associated to irrigation perimeters which rely on groundwater.



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



**Table 1.** Consumptive rates for different water uses (source: Margat and Treyer, 2004)

| Sector | Consumptive rate |
|---|---|
| Collectivities | 15 % |
| Agriculture | 80 % |
| Industry | 5 % |
| Power plants | 1,5 % |



**Table 2.** Satisfied demands, under the objective of economic value maximisation and using one-point hedging (option $V^+H^+$)

| Basin | Satisfaction rate (%) | | | | | | | |
| | Demand (quantity) | | | | Value | | | |
| | Past | | Future | | Past | | Future | |
| | spA | spB | spA | spB | spA | spB | spA | spB |
|---|---|---|---|---|---|---|---|---|
| 9 | 2.9 | 2.9 | 0.1 | 0.1 | 9.2 | 9.2 | 0.5 | 0.5 |
| 13 | 88.3 | 88.3 | 52.6 | 52.3 | 99.8 | 99.8 | 65.5 | 65.4 |
| 20 | 53.3 | 53.1 | 11.7 | 11.7 | 53.3 | 53.1 | 32.1 | 32.1 |
| 28 | 100.0 | 100.0 | 9.0 | 9.0 | 100.0 | 100.0 | 31.3 | 31.3 |
| 33 | 8.0 | 8.0 | 1.5 | 1.5 | 15.4 | 15.3 | 5.6 | 5.5 |
| 35 | 72.4 | 71.3 | 84.4 | 84.0 | 99.7 | 99.7 | 99.9 | 99.9 |
| 1170 | 4.9 | 0.8 | 3.5 | 0.3 | 10.2 | 4.7 | 6.5 | 1.5 |
| 1171 | 11.0 | 7.8 | 15.3 | 11.3 | 22.6 | 20.4 | 28.2 | 21.9 |
| 1178 | 13.5 | 7.1 | 5.3 | 0.9 | 15.2 | 6.9 | 6.5 | 0.7 |
| 1181 | 5.5 | 5.5 | 1.9 | 2.0 | 9.3 | 9.3 | 5.8 | 5.9 |
| 1186 | 40.5 | 34.4 | 21.9 | 15.2 | 48.9 | 43.3 | 29.2 | 22.1 |
| 1189 | 16.4 | 16.4 | 9.4 | 9.4 | 24.0 | 24.0 | 2.4 | 2.4 |
| 1190 | 11.1 | 5.4 | 8.1 | 0.3 | 12.2 | 7.6 | 6.7 | 0.7 |
| 1191 | 60.3 | 60.1 | 42.3 | 38.1 | 72.8 | 72.6 | 56.0 | 50.2 |
| 1192 | 31.8 | 12.3 | 17.5 | 0.5 | 60.0 | 33.0 | 28.6 | 2.6 |

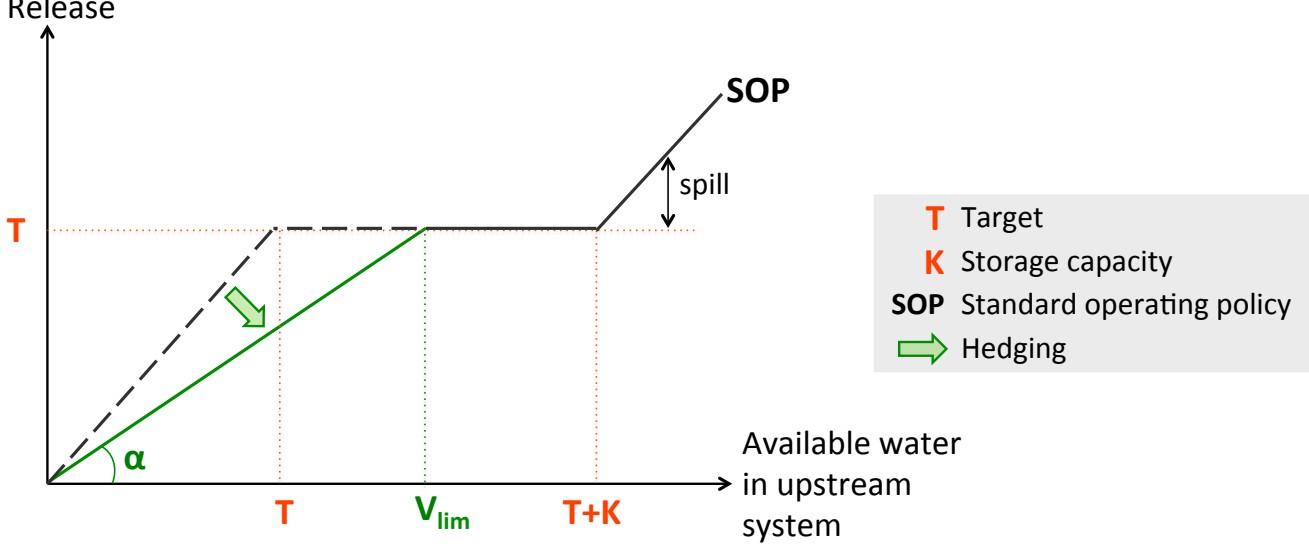

**Figure 1.** One-point hedging: one prudential parameter per reservoir ($\alpha$), rationing is initiated when the stored volume is lower than $V_{lim}$




**Table 3.** Change in demand satisfaction between the past period and the future period, under the objective of economic value maximisation and using one-point hedging (option $V^+H^+$)

| Basin | Change in satisfaction | | | |
| | satisfaction rate$_{future}$ - satisfaction rate$_{past}$ (in %) | | | |
| | Demand (quantity) | | Value | |
| | spA | spB | spA | spB |
|---|---|---|---|---|
| 9 | -2.7 | -2.7 | -8.7 | -8.7 |
| 13 | -35.7 | -36.0 | -34.2 | -34.4 |
| 20 | -41.6 | -41.4 | -21.1 | -21.0 |
| 28 | -91.0 | -91.0 | -68.7 | -68.7 |
| 33 | -6.4 | -6.5 | -9.7 | -9.8 |
| 35 | 11.9 | 12.6 | 0.2 | 0.3 |
| 1170 | -1.4 | -0.5 | -3.7 | -3.1 |
| 1171 | 4.3 | 3.5 | 5.6 | 1.6 |
| 1178 | -8.3 | -6.3 | -8.7 | -6.2 |
| 1181 | -3.6 | -3.5 | -3.5 | -3.4 |
| 1186 | -18.5 | -19.1 | -19.8 | -21.1 |
| 1189 | -7.0 | -7.0 | -21.6 | -21.6 |
| 1190 | -3.0 | -5.0 | -5.5 | -6.9 |
| 1191 | -18.0 | -22.0 | -16.8 | -22.4 |
| 1192 | -14.4 | -11.8 | -31.3 | -30.4 |

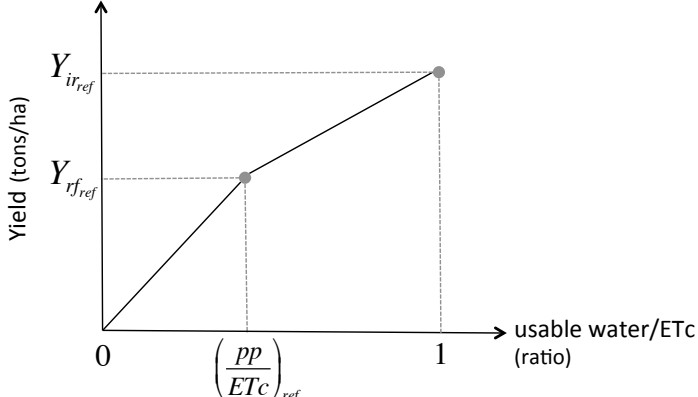

**Figure 2.** Modeling crop yield as a function of available water to ETc ratio. $Y_{ir_{ref}}$ and $Y_{rf_{ref}}$ are irrigated and rainfed crop yields of reference (in tons/ha), $ETc$ is the crop evapotranspiration, $pp$ the effective precipitation.




**Table 4.** Impact of demand prioritisation: difference in satisfaction with the option $V^+H^+$, compared to the results obtained with the option $V^-H^-$

| Basin | Difference in satisfaction | | | | | | | |
|---|---|---|---|---|---|---|---|---|
| | satisfaction rate$_{V+H+}$ - satisfaction rate$_{V-H-}$ (in %) | | | | | | | |
| | Demand (quantity) | | | | Value | | | |
| | Past | | Future | | Past | | Future | |
| | spA | spB | spA | spB | spA | spB | spA | spB |
| 9 | 1.5 | 1.5 | 0.1 | 0.1 | 7.6 | 7.6 | 0.3 | 0.3 |
| 13 | 0.0 | 0.0 | 0.4 | 0.3 | 0.0 | 0.0 | 11.1 | 11.4 |
| 20 | 2.5 | 2.5 | 5.5 | 5.5 | 2.4 | 2.4 | 20.2 | 20.2 |
| 28 | 0.0 | 0.0 | 3.5 | 3.5 | 0.0 | 0.0 | 22.5 | 22.5 |
| 33 | 2.7 | 2.7 | 0.7 | 0.6 | 7.3 | 7.2 | 3.7 | 3.6 |
| 35 | 0.0 | 0.0 | 0.0 | 0.0 | 0.0 | 0.0 | 0.0 | 0.0 |
| 1170 | -1.4 | 0.1 | -0.4 | 0.0 | 3.2 | 2.8 | 1.7 | 0.9 |
| 1171 | 0.1 | 0.2 | -0.4 | 0.3 | 9.0 | 9.2 | 10.1 | 7.2 |
| 1178 | -3.3 | -0.4 | 1.6 | -0.3 | 1.4 | 1.8 | 3.4 | 0.3 |
| 1181 | -0.8 | -0.8 | 0.2 | 0.3 | 4.1 | 4.1 | 4.3 | 4.4 |
| 1186 | -0.1 | -0.2 | 0.6 | 0.4 | 6.4 | 6.7 | 6.3 | 5.9 |
| 1189 | 1.6 | 1.6 | 0.7 | 0.7 | 16.6 | 16.6 | 1.7 | 1.7 |
| 1190 | 0.8 | 0.8 | 0.1 | 0.2 | 4.6 | 3.4 | 0.5 | 0.3 |
| 1191 | 2.0 | 1.9 | 0.8 | 1.5 | 11.3 | 11.2 | 6.4 | 6.0 |
| 1192 | -5.3 | 4.6 | 8.4 | -0.0 | 27.1 | 22.6 | 17.7 | 0.8 |

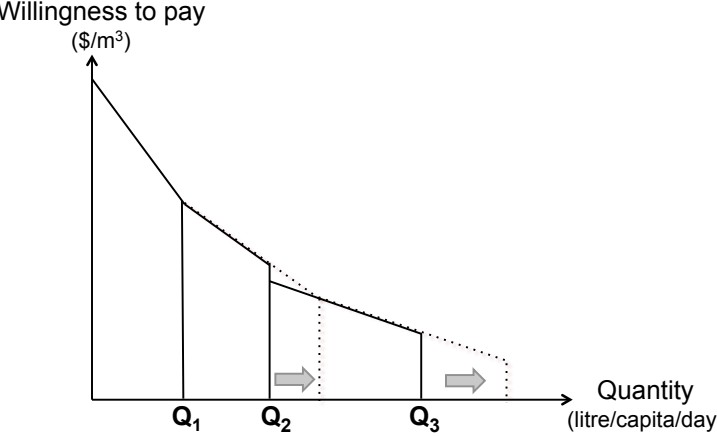

**Figure 3.** Domestic water demand function: economic development effect. $Q_1$, $Q_2$ and $Q_3$ are the volume limits of the three demand parts. The grey arrows represent the effect of economic development, which leads to larger demand by expanding the width of the blocks.





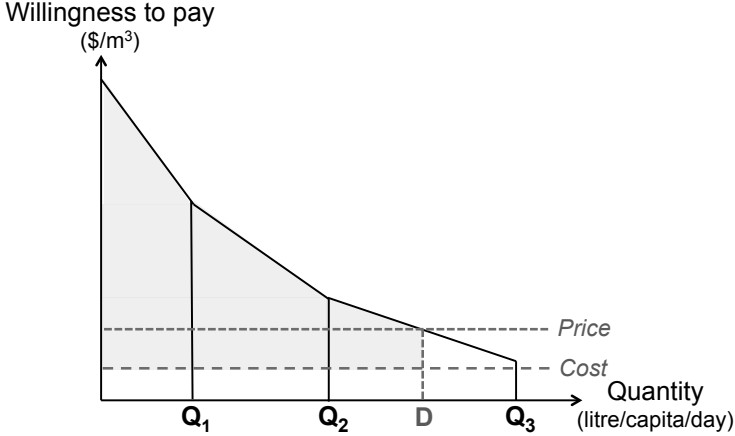

**Figure 4.** Domestic water demand function: surplus. $D$ is the level of demand corresponding to the level of price. The grey-coloured area under the curve represents total economic surplus.

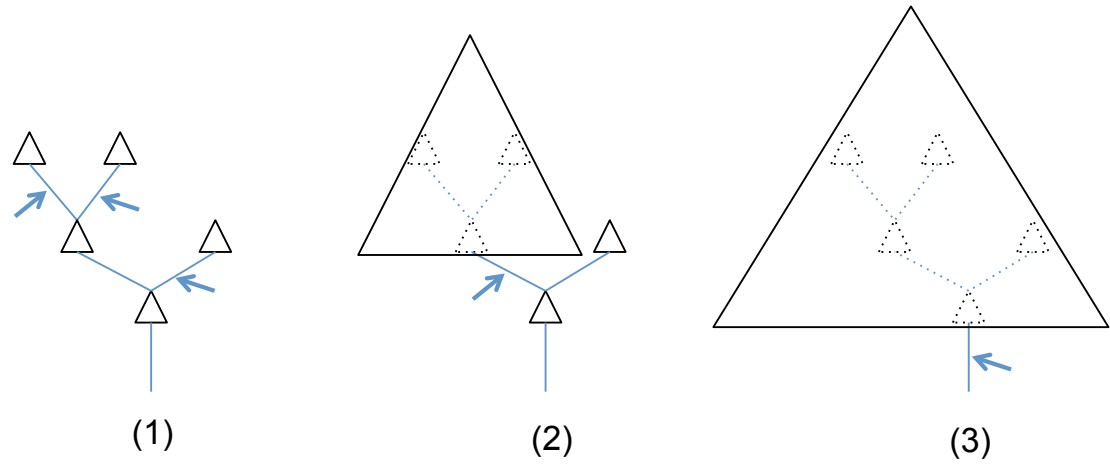

**Figure 5.** Satisfying demands of each stream: downwards tree traversal, with aggregations. (1) start with the most upriver streams, (2) aggregate upstream reservoirs' system when moving down, (3) repeat until reaching the root of the network.



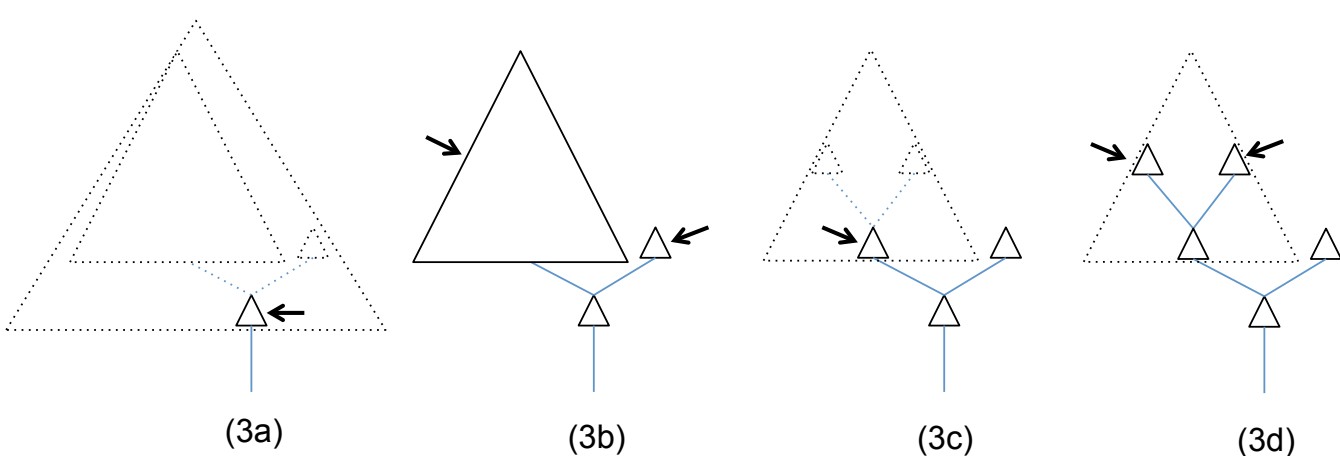

**Figure 6.** Distributing release between upstream reservoirs: upwards tree traversal, with disaggregations. Steps for case (3) of Figure 5







**Figure 7.** Map of Algerian reservoirs systems. The white area is the Mediterranean sea. Basins borders are in black. Light grey basins are basins without reservoirs. White triangles are reservoirs, and white lines are the upstream-downstream links between reservoirs. Numbered labels are located at the downstream root of each system.