# Peer review of "Large-scale water scarcity assessment under global changes: insights from a hydroeconomic framework"

_Hydrology and Earth System Sciences, 2015_

## Referee Comment (RC1) · Anonymous Referee #1 · 29 Mar 2016

The title of the paper is closely related to water scarcity assessment; this is also a reason why I'm very interested in the topic. However, after reading through the manuscript, it seems the research is almost nothing related to water scarcity assessment. It deals with the operation of reservoirs.

There are several graphs, i.e. Fig. 1-7. But no any graph really shows the results of the present paper. They either provide theoretical background or location of reservoirs.

For the tables, I also feel difficult to understand the results. For example, for Table 1, the sum of four categories is higher than 100%. For Table 2, for many Basins, the results are the same for scenario spA and spB (also for Table 3 and 4).

[Figure]

One issue is the validation of the models used in the paper. There is no calibration or validation presented. Hence, it is difficult to know the accuracy of the results. There are too many assumptions in the entire paper, and it is difficult to know the accuracy of results.

---

## Referee Comment (RC2) · Anonymous Referee #2 · 10 May 2016

Summary

This paper presents a generic hydroeconomic modeling framework for basin-scale water allocation and demonstrates its use for a case study in Algeria. As pointed out by the authors, many global-scale water resources models neglect inter-temporal trade-offs due to water storage in regulated multi-reservoir systems, which are important in many river basins. The strength of the approach presented here is that it relies on simple input datasets, which are available worldwide, but still captures economic trade-offs due to multi-reservoir storage. The weakness is a lack of validation and insufficient justification of the various simplifications for specific use cases. I have a number of review comments and details that could be addressed in a revised version of the manuscript.

[Figure]

Review Comments

1. Section 2.1, estimating water availability today and in the future: Some more details should be presented here. How good is the climate model in predicting present-day runoff in the region? Is 2050 runoff taken directly from the model or do you use some kind of change factor methodology? What is your assessment of the uncertainty of 2050 water availability? In order to compute future irrigation demands, do you directly use the simulated precipitation from the climate model? How good is the climate model in terms of predicting present precipitation? It would be good to briefly revise the main assumptions, limitations and sources of uncertainty and then refer to Portoghese et al. 2013 for details.

2. It is not entirely clear how the link between water allocation and agricultural yield is simulated: Figure 2 shows a piecewise linear relationship. Does this relationship apply at the time scale of the entire growing season or for individual growth stages? Irrigation agriculture presents the well-known problem of "delayed yields", i.e. the yield is a function of shortages occurring in all growth stages and shortages in one stage cannot be offset by surpluses in the next stage. Does the framework take this into account or is there a constant water value throughout the season, independent of irrigation history?

3. Section 3, network construction: I do understand the rationale behind the chosen approach, i.e. generating the network topology purely from the elevation model. It is attractive because you can generate a model without detailed knowledge about the system, but it is also dangerous, because many links that are outlined by the algorithm may not be there in physical reality and others, that the algorithm cannot find (e.g. South to North Water transfer in China. . .) may be present in reality. However, network topology to a large extent determines spatial and temporal trade-offs. I believe the authors should present more information to validate the network construction algorithm and to elucidate its limitations. If this is used on a new area, how can one establish trust in the outlined network and how can the network be validated?

4. Section 4, reservoir regulation: The optimization problem is solved using a GA. It would be good to report more details on the GA setup: Which are the decision variables (how many are there)? Is it the alpha and beta parameters? What was the computational effort, how was convergence etc.

5. Section 4.3.4 on tree traversal and also the corresponding appendix D are very short. A minimum amount of information should be given enabling the reader to understand how this works. Figs 5 and 6 do not communicate very well, captions need to be expanded.

6. As with all studies using complex modelling chains, uncertainty assessment is a real challenge here. How robust are the headline results reported in tables 2-4? Which of the reported differences are statistically different from zero? What is the largest contribution to uncertainty – future climate or economic valuation? No attempt is made in the paper to address the uncertainty of results. I know it is difficult, but authors must at least discuss the issue qualitatively, quantitative estimates would be much better.

Details

1. P1, Line3: "can they" should be "they can"

2. P2L1: "water use sectors" is more common

3. P3L1: It is not clear what is meant with "mostly quantitative" here. Why is this a limitation of such studies?

4. P6L9: "reservoir system" should replace "reservoirs' system"

5. P6L11: "valuation" is more common than "valorisation"

6. P7L4: "at best" should be "optimally"

7. Figure 7 should be much improved. Make an inset map showing the location of the area on the planet. Put a scale/coordinate system. Maybe use elevation model as background...

8. I believe figs 3 and 4 can be combined into one. Also, from the discussion given in appendix B, it seems that the demand functions should be piecewise horizontal, not piecewise linear. . .

---

## Author Comment (AC1) · 29 Jun 2016

Please find below our replies to Anonymous Referee #1 comments.

1. Water scarcity: "The title of the paper is closely related to water scarcity assessment; this is also a reason why I'm very interested in the topic. However, after reading through the manuscript, it seems the research is almost nothing related to water scarcity assessment. It deals with the operation of reservoirs."

> This paper looks at water scarcity, defined as the lack of sufficient water for the water uses. The developed model enables to assess demand satisfaction ratios, which are a measure of water scarcity, by assessing and comparing water demand and water

supply. The paper deals with reservoirs operation because reservoirs "help regulate climatic variability in time and space, to distribute water when needed by demands" (Cf. p.2 l.4-5, Introduction section) and thereby help mitigate water scarcity. It is therefore important to take into account reservoirs and their operation when assessing water scarcity. To make this clearer to the reader we can add a sentence to the introductory paragraphs, with a definition of what we mean by water scarcity.

2. Figures: "There are several graphs, i.e. Fig. 1-7. But no any graph really shows the results of the present paper. They either provide theoretical background or location of reservoirs."

> Results of the study are displayed in tables, because we found it to be the clearest way to present the results. Graphs are here to illustrate other elements. If there are more precise indications of missing figures, we could add those.

3. Tables: "For the tables, I also feel difficult to understand the results. For example, for Table 1, the sum of four categories is higher than 100%. For Table 2, for many Basins, the results are the same for scenario spA and spB (also for Table 3 and 4)."

> Table 1 does not display results of the study but data from Margat and Treyer (2004). The data are not to be summed. There is one piece of information for each row (i.e. each type of use): the consumptive share of the water used by each sector. Cf. p.5 l.8: "Water demands are not entirely consumptive (Table 1), they generate return flows that may help satisfy downstream demands." For each sector the consumptive share plus the return flow share would be equal to 100%. We will add the definition of "consumptive rate" in the caption of Table 1.

> Scenarios spA and spB are defined in Section 5.1.2, first paragraph. They are implemented to test the sensitivity of the results to the choice of spill modelling (before vs. after allocating water to the demands). We will add a sentence to the paragraph to clarify this purpose. The results for both spill scenarios are provided in the results tables to look at this sensitivity. We will add in the first sentence of the Results section (Section 5.2, p.9 l.9) that both ways of modelling spill are presented in the results table to show the sensitivity of the results to the spill modelling. In the paragraph p.9 l.21-23: figures are in % points, and not %. It will be corrected. The paragraph will also be more developed, discussing the model sensitivity to the spill modelling option. Low and strong sensitivities to the spill option will be discussed. In most basins the model is not very sensitive to the spill scenario, but in some basins the two spill scenarios yield important differences in results (e.g. Table 2, basin 1192). In tables 3 and 4, the results that are presented consist in differences in satisfaction rates between two situations, which smoothens the impact of the spill option.

More generally, the Results section will be improved: basins numbers will be replaced with the name of the main river in each basin; results that are the most reliable/unreliable will be visually identified in the tables.

4. Validation: "One issue is the validation of the models used in the paper. There is no calibration or validation presented. Hence, it is difficult to know the accuracy of the results. There are too many assumptions in the entire paper, and it is difficult to know the accuracy of results."

> We are aware that the lack of validation is a weakness of our work. As mentioned, a proper validation of the framework is limited by the lack of data. For the municipal and irrigation water demands, the methodologies were validated to the best of our ability. It was discussed in two previous papers that focussed specifically on the modelling of municipal water demand (Neverre and Dumas, 2015) and irrigation water demand (Neverre and Dumas, 2016 - which was not cited because it was previously under revision, but is now published). We will add information about the sensitivity analysis and the validation efforts that were carried out, and refer to these previous papers. For the reconstruction of demand-supply networks, the validation experiment carried out in Algeria is presented in Appendix F. It will be further discussed in the main text. For the reservoirs operation rules, unfortunately there is no adequate data for a validation (Cf. paragraph p.10 l.31). What we can do with the available data is perform a simple evaluation at the country scale in historical conditions (aggregating the different basins). This evaluation will be added in the revised version of the manuscript. However, this evaluation will have serious limitations: uncertainties about the data, incomplete or different coverage (not all supply sources/all demands included in the data/in the modelling framework).

> We had to make assumptions when precise data were not available. We tried to maintain the trade-off between relevance and precision. We tried to be transparent about the assumptions made and how it would affect accuracy. The objective was that the considered assumption would improve the modelling, compared with not taking the considered element into account.

> Most results presented in the present paper consist in differences between two situations: changes in demand satisfaction rates (indicator of water scarcity) between historical and future conditions, or changes in demand satisfaction rates between different modelling scenarios (with/without operation rules based on economic criteria). Even if there are biases in the magnitude of the modelling framework's results, presenting the results in terms of difference rather than absolute values is expected to offset magnitude biases.

For further details, please also see our replies to Anonymous Referee #2 comments: comment #1-iii and comment #6.

---

## Author Comment (AC2) · 30 Jun 2016

Please find below our replies to Anonymous Referee #2 comments. Original comments from the referee are in quotes and our replies follow (>).

—

1. Estimating water availability:

"Some more details should be presented here. How good is the climate model in predicting present-day runoff in the region? Is 2050 runoff taken directly from the model or do you use some kind of change factor methodology? What is your assessment of the uncertainty of 2050 water availability? In order to compute future irrigation demands,

do you directly use the simulated precipitation from the climate model? How good is the climate model in terms of predicting present precipitation? It would be good to briefly revise the main assumptions, limitations and sources of uncertainty and then refer to Portoghese et al. 2013 for details."

> i) Climate model: The climate model used is pertinent in the context of an application of our framework to the Mediterranean because it uses a stretched-grid global climate model zoomed on the Mediterranean coupled with a high-resolution oceanic model of the Mediterranean (Dubois et al., 2012). We do not have much information on the model biases. From our interactions with the model developers, we gathered that there was no particular bias identified, but the model may underestimate extreme events (Dubois et al., 2012).

Since the climate model is a coupled model it is not possible to evaluate the modelled runoff results by using historical precipitation and temperature as inputs (the model uses radiative forcing as input, the rest is endogenous). To our knowledge, there was no experiment of using the model only for runoff computation, decoupled from the rest. It is not possible to compare the model's results to historical times series, it can only be compared to statistics.

The only published data source we have on outflows from Algerian reservoirs is Pérennès (1993), for reservoirs built before 1920. These reservoirs may have been modified since. What we can do is use this data to check that the modelled unregulated flows to these reservoirs are higher than the regulated outflows reported in Pérennès (1993) data.

> ii) Outputs used: Yes runoff outputs of the model are directly used to compute runoff to the reservoir. To compute irrigation demands, we use a linear formula to determine the share of precipitation that runs off from total precipitation (following the methodology used in Döll, 2002). This information will be added to the manuscript (section 2.2.1), and we will also reference another paper which focuses more on irrigation demand, that was previously under revision but is now published (Neverre and Dumas, 2016). We will also specify that, in the present paper, irrigation needs are computed considering that the length of the growing seasons remains unchanged under future climatic conditions.

> iii) Limitations and sources of uncertainty: We will add a paragraph to discuss this. Of course relying on only one climate model, under only one future climate scenario is a serious limitation. We feel that the present paper is not the place to perform an uncertainty analysis. It would be necessary to compare several models and forcing scenarios. It is not possible to add such an analysis to the present paper; it would require a separate paper. What we can do here is discuss the uncertainty associated with the use of one model and one climate scenario. We will add this discussion in the revised manuscript.

We will also state more clearly that the focus of the paper is a demonstration of what can be done with the hydroeconomic framework. It will have to be run under different scenarios and further evaluated. This first application makes it possible to test the assumption that reservoir operation rules based on economic criteria will become more relevant to mitigate water scarcity under global changes than they are under historical conditions. The first results presented in the present paper invalidate this assumption. We obtain heterogeneous results between basins, and in some basins the assumption is invalidated. Cf. section 5.2: "For some basins (9, 33, 1170, 1189, 1190, 1191 and 1192), the positive impact of prioritization is more pronounced under past conditions than under future conditions." Running different scenarios would inform on the robustness of these findings, but we can already show a first invalidation: the benefits of operating rules based on economic criteria are not unequivocally increased with global changes.

—

2. Agricultural yield and water availability:

"Figure 2 shows a piecewise linear relationship. Does this relationship apply at the time scale of the entire growing season or for individual growth stages? [...] is there a constant water value throughout the season, independent of irrigation history?"

> Yes the piecewise linear relationship applies for the entire growing season. And there is a constant water value throughout the season. It is an average value that is calculated over the whole climatology (50 years) of the considered time period (historical or future period).

"Irrigation agriculture presents the well-known problem of "delayed yields", i.e. the yield is a function of shortages occurring in all growth stages and shortages in one stage cannot be offset by surpluses in the next stage."

> Irrigation needs are computed at a monthly time step, so there is no possibility of offsetting past shortages by surpluses in the next month: each month, water is allocated to the irrigation sector up to the crops needs; there are no irrigation surpluses.

> We could take into account the different phases of the growing season when estimating the loss due to irrigation water shortages over the year, but there would then be a discrepancy as the economic value we would then obtain would differ from the value that is used for the monthly allocation decisions. To take into account irrigation history, it would be necessary to have a dynamic model, in which water value for the crop would be re-evaluated each month depending on the water allocated during the previous months.

We do not try to capture this variability of the value within the season. We think that we do not need this level of precision here. The results we present in the paper are average results for the whole climatology (50 years). What we want is to have an evaluation of the average opportunity cost for irrigation, to be able to consider trade-offs between different water use sectors. When there is not enough water to satisfy irrigation needs, there is not only a loss for the irrigated culture, but implicitly there is a change of the cultural system from irrigated to rainfed. This switch is not modelled

explicitly on a yearly basis, rather the average difference of value gives an idea of the expected water value for the irrigation sector.

This will be made clearer in the manuscript, and we will reference another paper which focuses more on irrigation demand projection and valuation, that was previously under revision but is now published (Neverre and Dumas, 2016).

—

3. Network reconstruction:

"I do understand the rationale behind the chosen approach, i.e. generating the network topology purely from the elevation model. It is attractive because you can generate a model without detailed knowledge about the system, but it is also dangerous, because many links that are outlined by the algorithm may not be there in physical reality and others, that the algorithm cannot find (e.g. South to North Water transfer in China...) may be present in reality. However, network topology to a large extent determines spatial and temporal trade-offs. I believe the authors should present more information to validate the network construction algorithm and to elucidate its limitations."

> i) Water transfers: Indeed, this type of link between reservoirs cannot be generated automatically in the model. However, these links are infrequent enough to be added manually when they are known. The model is not yet able to handle two interconnected basins, since it cannot handle several downstream systems yet. This feature would have to be added to the framework. It will be the subject of future developments of the framework. Links between reservoirs and demands located in different basins are already considered in the framework, as explained in the paper.

> ii) Erroneous associations: Indeed, the algorithm may find links that are not present in reality. The validation of the network reconstruction in Algeria showed that such errors exist (Cf. Appendix F). This validation experiment on Algeria was the subject of a whole paper, which is currently under revision (Nassopoulos and Dumas, under

revision). The main results that were relevant for the present paper are presented in the Appendix F. We will further discuss this issue in the main text.

"If this is used on a new area, how can one establish trust in the outlined network and how can the network be validated?"

> iii) Application to another area: To validate the network reconstruction, it is necessary to have knowledge of real links, to compare them with the links that result from the algorithm. If an evaluation results in a change of the modelling, it could lead to general improvement in matching with observed data. However, in a general way, a validation cannot be transposed. Each time a model is applied to a new area, it is not possible to know a priori how valid it will be.

—

4. Genetic Algorithm:

"It would be good to report more details on the GA setup: Which are the decision variables (how many are there)? Is it the alpha and beta parameters? What was the computational effort, how was convergence etc."

> Yes, the decision parameters are the alpha and beta parameters. There is one alpha parameter for each tree node (i.e. 18 alpha parameters in total for basin #1186, or 4 for basin #1175 for instance), and one beta parameter for each parallel branch (i.e. 6 beta parameters in total for basin #1186, or 1 for basin 1175). We used a population size of 100 and 20 generations. We tried different numbers of runs and generations, and chose the best trade-off between computation time and convergence. We will add a paragraph in the manuscript to give more information about this.

—

5. Tree traversals:

"Section 4.3.4 on tree traversal and also the corresponding appendix D are very short.

[Figure]

A minimum amount of information should be given enabling the reader to under- stand how this works. Figs 5 and 6 do not communicate very well, captions need to be expanded."

> The text describing tree traversals and the captions of the associated figures will be expanded.

—

6. Uncertainty:

"As with all studies using complex modelling chains, uncertainty assessment is a real challenge here. How robust are the headline results reported in tables 2-4? Which of the reported differences are statistically different from zero? What is the largest contribution to uncertainty – future climate or economic valuation? No attempt is made in the paper to address the uncertainty of results. I know it is difficult, but authors must at least discuss the issue qualitatively, quantitative estimates would be much better."

> We will add some discussion in the manuscript (Cf. also comment 1.).

> We will add a qualitative/semi quantitative evaluation of the introduction of economic valuation in the framework. We will show separate results for the domestic and irrigation sectors. These results show that we obtain more sensible results in terms of demand satisfaction when economic rules are used than without economic rules: with economic rules the satisfaction of domestic demand increases to the detriment of irrigation demand, and the demand satisfaction rates obtained are closer to expected figures (except for 1 basin, where domestic demand satisfaction is unexpectedly low).

For further details, please also see our replies to Anonymous Referee #1 comments: comment #4.

—

Details:

Thank you for the suggested corrections, we will take all of them into account. Please see below our answers to some specific comments.

3. "P3L1: It is not clear what is meant with "mostly quantitative" here. Why is this a limitation of such studies?"

> The sentence will be rephrased. We meant that these studies focus on the quantity of water, without incorporating an economic assessment (no economic value of water, only quantities).

7. "Figure 7 should be much improved. Make an inset map showing the location of the area on the planet. Put a scale/coordinate system. Maybe use elevation model as background."

> Figure 7 will be improved as suggested. We will keep the map in grayscale, as the aim of using grayscale is to improve accessibility for colour-blind readers.

8. "I believe figs 3 and 4 can be combined into one. Also, from the discussion given in appendix B, it seems that the demand functions should be piecewise horizontal, not piecewise linear."

> Figures 3 and 4 will be combined.

> The domestic demand function is piecewise linear, with slopes (Cf. text in Appendix B and Figures). You may have been thinking of piecewise horizontal because of section 4.2: In this section, we explain how projected demands are broken down and grouped into classes based on their value. So the domestic demand function is piecewise linear, but then projected demands are discretized to ease operating rules determination, at the cost of using an approximation of the domestic demand function value.

—

References:

Dubois, C. et al. (2012) Future projections of the surface heat and water budgets of

the Mediterranean Sea in an ensemble of coupled atmosphere–ocean regional climate models. Climate Dynamics (39).

Nassopoulos, H. and Dumas, P. (under revision) Reconstructing river basin anthropogenic networks with a global coverage: A validation on Algeria.

Neverre, N. and Dumas, P. (2016) Projecting Basin-Scale Distributed Irrigation and Domestic Water Demands and Values: A Generic Method for Large-Scale Modeling. Water Economics and Policy, Vol. 2, No. 3.

Pérennès, J-J. (1993) L'eau et les hommes au Maghreb. Contribution à une politique de l'eau en Méditerranée. Paris, Karthala. pp. 646.
* * *